# Effectiveness of Global Postural Re-Education in Chronic Non-Specific Low Back Pain: Systematic Review and Meta-Analysis

**DOI:** 10.3390/jcm10225327

**Published:** 2021-11-16

**Authors:** Gloria Gonzalez-Medina, Veronica Perez-Cabezas, Carmen Ruiz-Molinero, Gema Chamorro-Moriana, Jose Jesus Jimenez-Rejano, Alejandro Galán-Mercant

**Affiliations:** 1Department Nursing and Physiotherapy, Faculty of Nursing and Physiotherapy, University of Cadiz, 11009 Cadiz, Spain; gloriagonzalez.medina@uca.es (G.G.-M.); carmen.ruizmolinero@uca.es (C.R.-M.); alejandro.galan@uca.es (A.G.-M.); 2Research Group: CTS-986 Physical Therapy and Health (FISA), Institute of Research in Social Sustainable Development (INDESS), 11009 Cadiz, Spain; 3Biomedical Research and Innovation Institute of Cadiz (INiBICA), Research Group: [CTS1038] eMpOwering Health by Physical Activity, Exercise and Nutrition (MOVEIT), Research Unit, Puerta del Mar University Hospital, 11009 Cadiz, Spain; 4Department of Physiotherapy, University of Seville, 41009 Seville, Spain; gchamorro@us.es (G.C.-M.); jjjimenez@us.es (J.J.J.-R.); 5Research Group [CTS305] “Area of Physiotherapy CTS305”, University of Seville, 41009 Seville, Spain

**Keywords:** chronic low back pain, global postural re-education, disability, physical therapy, pain

## Abstract

Background: The aim of this systematic review and meta-analysis was to evaluate the global postural re-education (GPR) program’s effectiveness compared to other exercise programs in subjects with persistent chronic low back pain. Methods: A systematic review and meta-analysis were carried out using PRISMA2020. An electronic search of scientific databases was performed from their inception to January 2021. Randomized controlled trials that analyzed pain and patient-reported outcomes were included in this review. Four meta-analyses were performed. The outcomes analyzed were disability due to back pain and pain. The risk of bias and quality of evidence were evaluated. The final search was conducted in March. Results: Seven trials were included, totaling 334 patients. The results showed improvement in pain measured by Visual Analogue Scale (VAS) (Standardised Mean Difference (SMD) = −0.69; 95% Confidence Interval (CI), −1.01 to −0.37; *p* < 0.0001), Numerical Pain Scale (NRS) (SMD = −0.40; 95% CI, −0.87 to 0.06); *p* = 0.022), VAS + NRS (SMD = −1.32; 95% CI, −1.87 to −0.77; *p* < 0.0001) and function (Roland Morris Disability Questionnaire (RMDQ)) (SMD = −0.55; 95% CI, −0.83 to −0.27; *p* < 0.0001) after GPR treatment. Conclusion: This meta-analysis provides reliable evidence that GPR may be an effective method for treating LBP by decreasing pain and improving function, with strong evidence.

## 1. Introduction

Global postural re-education (GPR) is a physical therapy method, developed by Phillipe Souchard in the 1950s. This therapeutic approach is based on an integrated idea of the muscular system as formed by muscle chains, which can be shortened resulting from constitutional, behavioral and psychological factors [1]. Physical therapists use exercise to eccentrically stretch muscle chains. For this purpose, a series of active gentle movements and postures was aimed at realigning joints, stretching shortened muscles and enhancing the contraction of antagonist muscle. The program implies an active involvement of the patient [2]. This physical therapy modality has been used to improve the health status of patients with various pathologies [2,3,4,5,6], although the available studies do not provide sufficient evidence to draw firm conclusions.

This study focused on the analysis of GPR efficacy for chronic low back pain (LBP) patients, since we found several authors who have studied this topic [7,8,9]. The LBP therapeutic approach for physical therapists should be a priority, since, according to the World Health Organization [10], it is not only a musculoskeletal health problem, but also involves other dimensions of the individual. In addition, is one of the most common health problems worldwide [3,4]. As described by Popescu and Lee [11], its prevalence is very high, reaching 84% in adults. It is the most common musculoskeletal complaint in the emergency department and 2% of these patients require hospitalization [12]. The causes of chronic LBP are numerous, including many musculoskeletal, inflammatory, malignant, or visceral problems [10,11,12]. According to previous studies, risk factors are related to being female, being older and having a family history or a personal history of LBP [13].

The most common chronic LBP treatments are pharmacological, minimally invasive interventional therapy and rehabilitation, within which exercise is recommended [14]. Previous reviews have been conducted on the effectiveness of different therapies in the LBP [7,15,16]; however, none of them focus on the effectiveness of RPG in the chronic LBP.

Because of this, the objective of this study was to carry out a systematic review and meta-analyses of the effectiveness of global postural re-education in chronic non-specific low back pain.

## 2. Materials and Methods

### 2.1. Study Design

Following the criteria of the PRISMA statement [17], a systematic review and meta-analysis was carried out. The study has been registered in the PROSPERO platform in December 2019 (CRD42020161157). Bibliographic searches were conducted between 4 December 2019 and 12 January 2020. A final search in March was conducted before the manuscript was finally finalized. Searches were carried out using PubMed, Physiotherapy Database (PEDro), Scientific Electronic Library Online (SciELO) and Web of Science (WoS) databases.

### 2.2. Search Strategy

The search strategy was: (“Pain”[Mesh] OR “Acute Pain”[Mesh] OR “Pelvic Girdle Pain”[Mesh] OR “Musculoskeletal Pain”[Mesh] OR “Chronic Pain”[Mesh] OR “Visceral Pain”[Mesh] OR “Nociceptive Pain”[Mesh] OR “Pain Perception”[Mesh] OR “Pain, Referred”[Mesh] OR “Shoulder Pain”[Mesh] OR “Neck Pain”[Mesh] OR “Pelvic Pain”[Mesh]) AND “Global Postural Reeducation”; (“Range of Motion, Articular”[Mesh]) AND “Global Postural Reeducation); (“Quality of Life/psychology”[Majr]) AND (Global Postural Reeducation); (“Urogenital System”[Mesh] OR “Male Urogenital Diseases”[Mesh] OR “Female Urogenital Diseases”[Mesh] OR “Urogenital Surgical Procedures”[Mesh] OR “Pelvic Organ Prolapse”[Mesh]) AND (Global Postural Reeducation); Global Postural Reeducation.

Filters were used to include studies conducted in humans; age of participants: over 18 years; type of study: clinical trials and reviews. We excluded those published in Korean, clinical trials without control groups, duplicates and those in which the main pathology is not chronic LBP. The flowchart describes the process of obtaining the results, selection and eligibility.

### 2.3. Criteria for Considering Studies for This Review

The eligibility criteria were clinical trials and studies conducted in people over 18 years old with chronic non-specific low back pain [18]. The main intervention was global postural re-education. The comparisons were isostretching [9,19], standard chronic LBP protocol [20,21], back school exercises [22], stabilization exercises [8] and drug treatment [23].

The outcome measures considered were level disability perception due to back pain using the Roland–Morris Disability Questionnaire (RMDQ) and Oswestry Disability Questionnaire (ODI). For pain measure, the Visual Analogue Scale (VAS) and the Numeric Rating Scale (NSR) were used. In addition, other outcome measures were the fingertip-to-floor test (FFT), the Short-Form Health Survey (SF-36), the Borg Scale, the Beck Inventory and Range of Articular Motion (ROM).

The grey literature was reviewed. The studies considered important for this research were requested from the main authors. The studies were selected by two blinded and independent researchers. For the final selection, a discussion was held to determine the outcome. In cases of disagreement, the help of a third investigator was sought who determined, through a vote, whether or not to include the study. The Mendeley platform was used to register the studies included.

### 2.4. Data Extraction

Data extracted from clinical trials were: bibliography (authors, journal, year of publication, database(s)); study characteristics (topic, study aim, study design, sample size, groups—number of groups and size—dropout, randomized—yes/no); participant characteristics (gender, age, symptoms, characteristics); description of the intervention (intervention and control groups, number of treatment sessions, duration of each session, total treatment time); results assessed; study results and DOI. The information was obtained with an in-depth, unbiased reading by one of the researchers and verified by another. Only publicly available data were taken into account. Stratified data were managed with the Excel 2016 program (Microsoft Corporation, Redmond, WA, USA). The variables for which the data were sought were perceived level of disability and pain.

### 2.5. Data Analysis and Outcomes

The Review Manager (RevMan 5.3) software version (Cochrane IMS) [24] was used for the calculation of the meta-analysis, forest plot and funnel plot. Epidat software version 3.1 (Public Health Information Service of the General Direction of Public Health of the Regional Ministry of Health (Xunta de Galicia) and the Health Analysis and Health Information Systems Unit of the Health Information Systems of the Panamerican Health Organisation (OPS-OMS), Santiago de Compostela, Spain; Washington DC, USA) [25] was used to calculate the risk of publication bias and sensitivity.

Subgroup calculation was used for the meta-analysis. The generic inverse of variance with standardized mean difference was applied. Scales measuring the same variable were associated in these subgroups. In all cases, the confidence interval (CI) was 95%.

The I^2^ coefficient was used to determine the degree of heterogeneity. More specifically, the following values were taken: I^2^ > 50% and/or Chi^2^ test (*p* < 0.05), indicating substantial heterogeneity where random effect models were applied and I^2^ < 50% and/or Chi^2^ test (*p* > 0.05), indicating substantial homogeneity where the fixed effect model was applied. For the calculation of the risk of publication bias, the Begg and Egger tests were applied where possible.

### 2.6. Evaluation of the Quality and Clinical Relevance

The Physiotheray Evidence Database (PEDro) scale [26] was applied to determine the quality of the studies. It is a tool to assess the internal validity of clinical trials and guide researchers in their decision making. It scores clinical trials with 11 items. Each item is scored with 0 or 1. The trials are scored with the sum of the scored items, up to a total of 10 points, since item 1 does not count in the overall sum. The higher the value of PEDro, the higher the internal validity. Two independent blinded researchers selected the studies. Subsequently, a discussion took place for the final selection and the arithmetic mean was used to obtain the final results.

The classification of the evidence was carried out with the Grading of Recommendations Assessment, Development and Evaluation (GRADEPro) [27]. It consists of the following five items: risk of bias, inconsistency, indirectness, imprecision and other considerations. Each domain was defined as not serious, serious, or very serious. The resulting quality assessment of the evidence was classified as high, moderate, low, or very low.

## 3. Results

The number of studies screened, assessed for eligibility and included in the review with reasons for exclusions is shown in the flow chart (Figure 1).

### 3.1. Data Extraction

The selected studies were published between 2010 and 2016, with 2015 being the year with the most studies published on this subject. The main objective of the studies included was to test the effectiveness of RPG in the chronic LBP. For this purpose, RCTs were developed. The sample size for all studies was 357 participants. Three studies reported the loss of patients [8,9,20]. A total of 45 subjects did not complete the trial. The sample was mainly divided into two groups [8,9,20,21,23], although in some studies, there were three groups [8,19,22]. All group allocation was random with the exception of two studies [8,20].

Regarding the gender of the subjects, in all the studies, the number of females was greater than that of males. However, the proportion varies according to the study. In some, the number of female participants was four times that of male participants [20,23]. In others, the number of participants by sex was more equal [8,19]. In relation to the age of the participants, there was homogeneity among all the authors, with an average age of 49.73–60.4 years [8,9,20,21,22,23], with the exception of Guimarães [19] who applied an age range between 19 and 60 years.

The main intervention in all studies was GPR. The comparisons were isostretching [9,19], standard chronic LBP protocol [20,21], back school exercises [22], stabilization exercises [8] and drug treatment [23]. The number of GPR sessions varied between ten [8,21,22], twelve [9,19] and fifteen [20] sessions. Their duration ranged from thirty [21], forty [22] and forty-five [9] minutes to one hour [8,19,20] (see Table 1).

### 3.2. Data Analysis and Meta-Analyses

After the study selection process, two articles were selected for the qualitative study and five for the quantitative study. A total of three meta-analyses were performed. Disability due to back pain using the Roland–Morris Disability Questionnaire (RMDQ) and Oswestry Disability Questionnaire (ODI), pain by the Visual Analogue Scale (VAS) and the Numeric Rating Scale (NSR) and quality of life by the SF36 questionnaire.

In addition, other outcomes measured have been found such as the fingertip-to-floor test (FFT), Borg Scale, Beck Inventory and Range of Articular Motion (ROM). In these cases, they were only used once, so a meta-analysis could not be performed.

Figure 2 demonstrates the effects of GPR on disability. It was suggested that GPR could significantly improve the perceived level of disability, with no heterogeneity (*p* < 0.0001, I^2^ = 0%). A small effect (SMD = −0.49, CI 95%: −0.70 to −0.27) was found in the intervention groups using GPR compared to other treatments/control.

A meta-analysis by subgroups was performed for the pain variable, since different outcome measures were included: VAS, NSR and SF36 pain (Figure 3). Regarding the effect of the RPG on pain, it was observed to significantly improve with heterogeneity (*p* = 0.03, I^2^ = 55%). A small effect (SMD = −0.38, CI 95%: −0.72 to −0.04) was found in the intervention groups using GPR compared to other treatments/control.

Nevertheless, the effect of GPR on quality of life became non-significant (*p* = 0.90), with no heterogeneity but close to it (I^2^ = 49%) (Figure 4).

### 3.3. Risk of Bias, Sensitivity and Heterogeneity

The results of the Begg and Egger tests (*p* > 0.05) (Table 2) indicate that there is no publication bias. The funnel diagram corroborates this information (Figure 5). On the other hand, the sensitivity analysis indicated that none of the studies included in this meta-analysis substantially modified the overall results when eliminated. In the case of the ODI and NRS variables, neither the Egger test nor the sensitivity analysis could be performed because only two studies were included in these meta-analyses. This is also shown in the following funnel plots (Figure 5). The sensitivity analysis indicated that no study substantially modified the overall results when eliminated.

### 3.4. Evaluation of Clinical Relevance

The results for the quality of the studies, measured with the PEDro scale, are shown in Table 3. In this case, the quality of all the selected RCTs was measured, regardless of whether they were included in the meta-analysis or not.

The quality of studies, in general, is average. The quality of the studies included in the meta-analysis is medium-high, with the exception of the study of Guimarães, M.L. et al. [19], which is very low.

The items that were least met in the methodology of these studies were: item 3 (allocation was concealed); item 5 (there was blinding of all subjects) and item 6 (there was blinding of all therapists who administered the therapy), those who did not complete any of the studies; and item 9 (intention to treat).

The quality of the evidence, as measured by GRADEPro, is shown in Table 4. The quality of the evidence is moderately high.

## 4. Discussion

The objective of this study was to determine the efficacy of global postural re-education in chronic low back pain. Taking into account our results, this meta-analysis suggests that global postural re-education is effective as a treatment in adults diagnosed with chronic LBP in terms of perceived level of disability and pain reduction. These results agree with previous studies, which have shown the effectiveness of GPR treatment [11]. In other pathologies, GPR acts on muscle flexibility [9,12,13,14,15,16,28,29], postural organization [3,29,30], functionality [6,20,23], quality of life [6,15,21,30,31], reducing pain [3,15,16,19], fatigue [23] and others [20,22,32,33,34].

There are many measurement tools to assess low back pain/disorders and their ability has been demonstrated [15,35,36,37]. The RMQ was used in five of the studies included in this review [8,9,20,21,23]. This questionnaire is a gold standard for the measure of perceived level of disability in chronic LBP [38,39]. The ODI scale was used in two studies [8,23]. These scales are considered the gold standard for measuring disability and quality of life (QoL) impairment for adults with chronic LBP [40]. Likewise, pain was measured with validated scales such as VAS [8,19,23], NRS [20,21] and SF36 [19,23]. The quality of life variable was measured with the SF36 questionnaire [19,23]. This fact made the development of the meta-analysis possible. The statistical analysis could not be performed with the other measurements performed (fingertip-to-floor test [8], sit and reach test [9], dynamometry [9], ultrasound examination [21], Beck Inventory [23], postural analysis [9], Borg Scale CR10 [23] and goniometry [23].

A meta-analysis by subgroups was carried out for the disability variable, since different outcome measures were included: RMDQ and ODI. The meta-analysis showed significant results with a small effect in favor of the GPR intervention for the perceived level of disability measured with RMDQ and ODI and pain reduction. These results are important, since all the objectives proposed for the clinical significance of these variables are met [36,37,41]. In the case of RMDQ, both the range of improvement (from 0 to 20 points) [42] and the percentage of improvement (30%) were met [37]. Other authors who analyzed interventions in chronic LBP, i.e., yoga [43], obtained a lower SMD value (SMD = −0.30, 95% CI = −0.51 to −0.10, *p* = 0.003, I^2^ = 0%) than that found in the present study with the RMDQ.

The meta-analysis by subgroups was performed for the pain variable, since different outcome measures were included: VAS, NSR and SF36 pain. Regarding the effect of the RPG on pain, it was observed to significantly improve with heterogeneity (*p* = 0.03, I^2^ = 55%). A small effect (SMD = −0.38, CI 95%: −0.72 to −0.04) was found in the intervention groups using GPR compared to other treatments/control. These results coincide with Lomas et al.’s study [44], in which the effects of GPR for the treatment of spinal disorders on pain were studied (SMD = −0.63; 95% CI, −0.43 to −0.83). However, in other types of diseases of rheumatic origin such as ankylosing spondylitis, GPR did not show significant results in reducing pain [45]. However, the included studies did not have enough evidence to perform meta-analysis segregating pain in the short, medium and long term.

Nevertheless, the effect of GPR on quality of life became non-significant. The characteristics of the sample studied differ from other studies that analyzed chronic LBP. In this review, the number of women was greater than that of men. This fact is in line with previous studies that indicate that women are more affected with chronic LBP [46,47]. In relation to the age, there is homogeneity among all the authors [8,9,19,20,21,22,23]. Both options can be accepted, as the literature indicates that adults of working age are the most vulnerable group for low back pain worldwide [32].

The RPG interventions used in the studies included in this review were very similar. Two to three postures were used, mainly focused on the treatment of the posterior chain [8,9,19,21,22,23]. Castagnoli [20] applied the posture depending on the muscular imbalance, but indicated which ones he used. The treatment time was around 30–60 min. The comparisons were isostretching [9,23], standard chronic LBP protocol [20,21], back school exercises [21], stabilization exercises [8] and drug treatment [23].

It should be noted that in the qualitative analysis of the studies, we observed that there were no significant differences between the groups when stretching exercises were applied in the comparison [9,19,20]. Studies whose comparison was isostretching [9,19] could not be included in the meta-analysis. An intervention study was included that contained some type of muscle stretching [20]. Likewise, another was included that did not specify the intervention and whose sample was very small [22]. These last two unsettle the meta-analysis results.

The isostretching exercise has principles in common with the GPR. In both cases, the same process, that is, viscoelastic stress relaxation, takes place and muscles are maintained in a static elongated position, regardless of the type of stretching [48]. This fact could justify the non-difference between the groups.

The overall quality of the studies was medium to high. Bonetti et al. [8] and Moreschi et al.’s [9] studies stand out as the ones that obtained the highest scores on the PEDro scale and Lawand’s study [23] for being the one with the lowest score. The difficulty in obtaining the maximum score on the PEDro scale was mainly because it is very complicated in this type of intervention to blind the patient and the physiotherapist: because, on the one hand, the patient has to perform the posture as correctly as possible and, on the other hand, the physiotherapist has to re-evaluate the treatment according to the patient’s evolution [1].

The important points of this review and meta-analysis are that the grey literature has been explored and a specific and in-depth study of the RPG in the LBP has been carried out based on the literature review and its statistical study. However, there are limitations such as the number of studies found, the heterogeneity in the use of measurement tools and the low quality of some of these. Therefore, we suggest that these points should be considered in further studies.

## 5. Conclusions

GPR is beneficial for chronic LBP in improving functional limitation and reducing pain perception. We suggest more future studies of good methodological quality to clarify the usefulness of RPG in other parameters such as the measurement of disability.

## Figures and Tables

**Figure 1 jcm-10-05327-f001:**
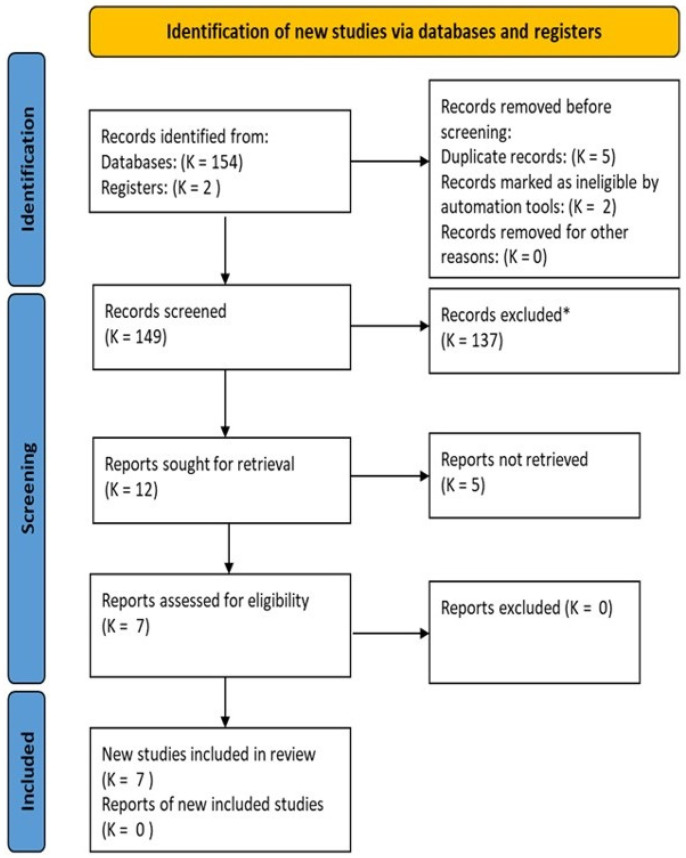
Flow chart.

**Figure 2 jcm-10-05327-f002:**
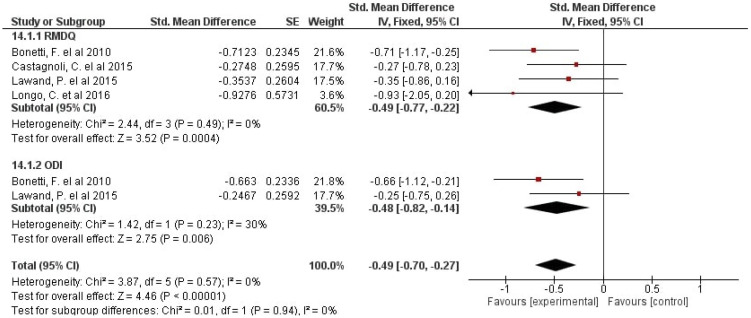
Effects of GPR on disability.

**Figure 3 jcm-10-05327-f003:**
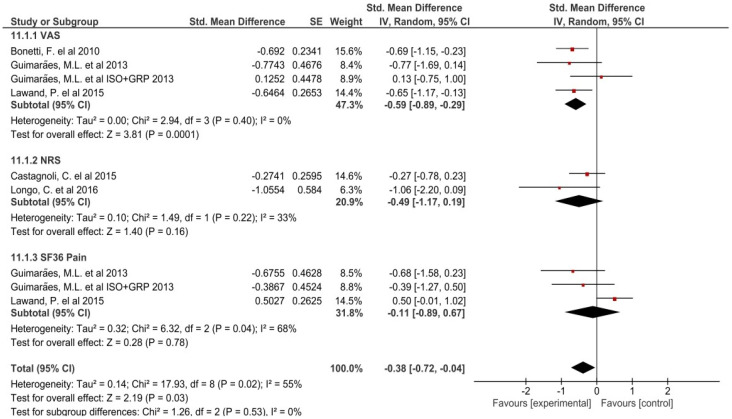
Effects by subgroups of GPR on pain.

**Figure 4 jcm-10-05327-f004:**
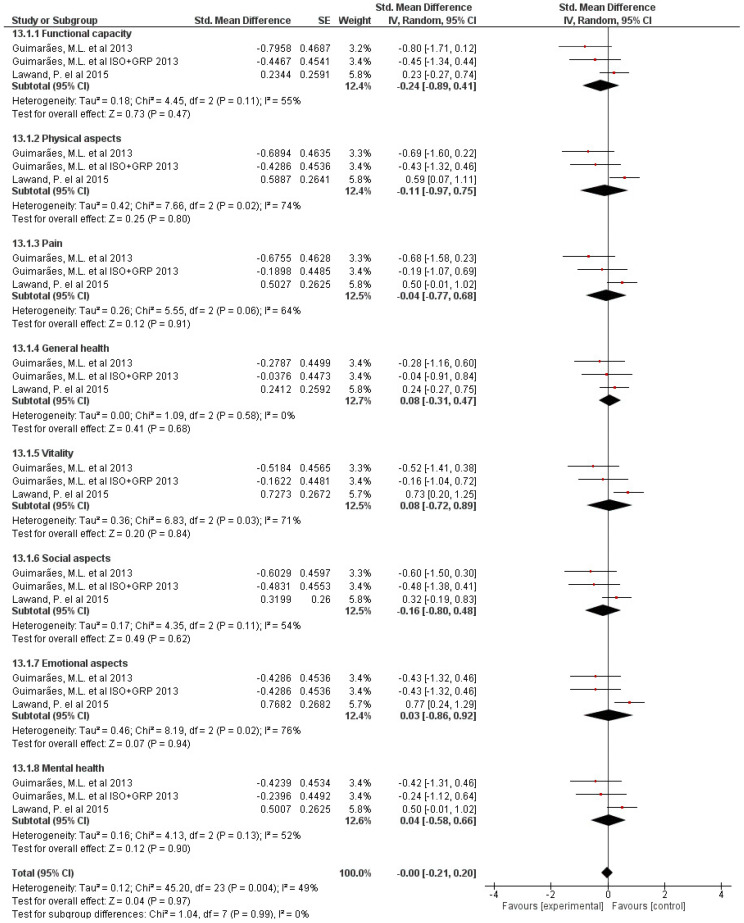
Effects of GPR on quality of life.

**Figure 5 jcm-10-05327-f005:**
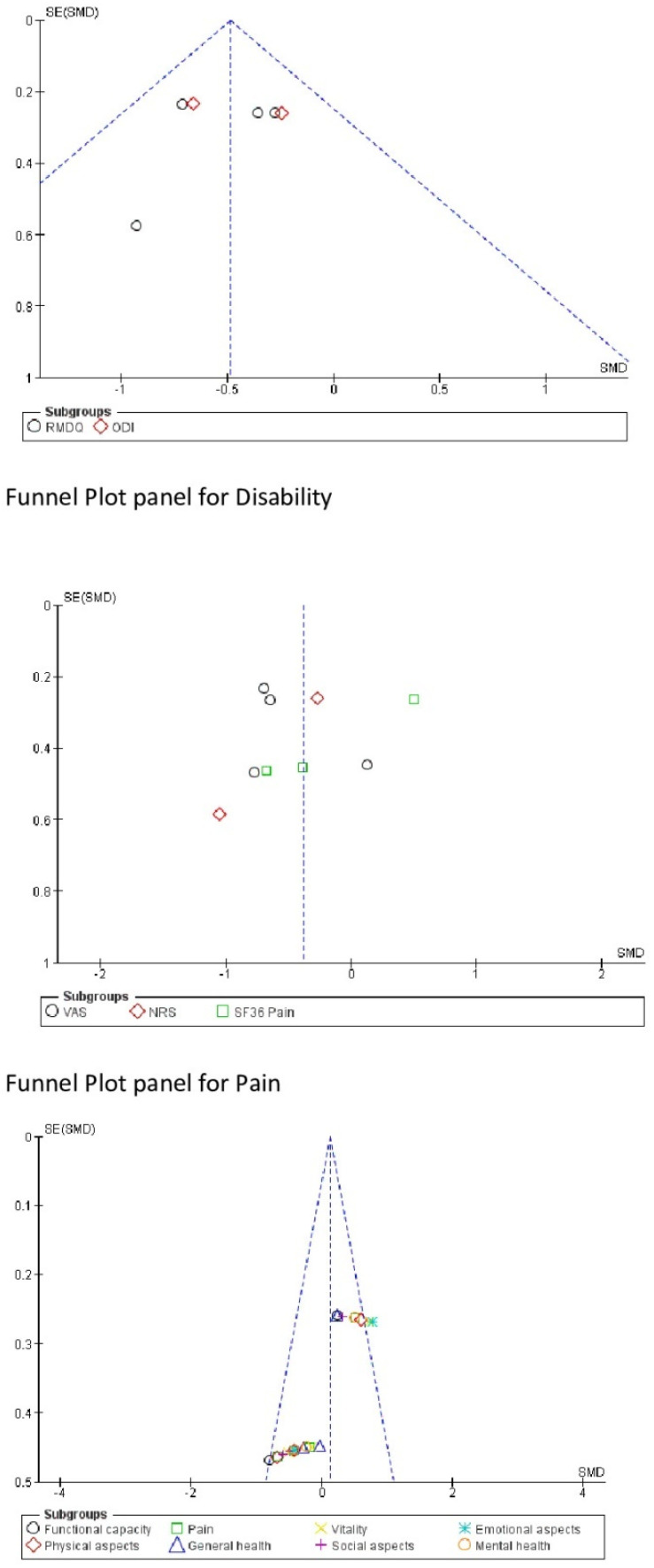
Publication bias.

**Table 1 jcm-10-05327-t001:** Characteristics of the studies.

The Authors Year/Objective	Sample	Intervention	Timeline and Follow-Up	Outcome Measure	Study Results
Bonetti, F. et al., 2010 [8] To evaluate the GPR effectiveness as compared to a Stabilization Exercise (SE) program in subjects with persistent low back pain (LBP) at short- and mid-term follow-up.	*n* = 100 Gender G1: M:22; F: 28 G2: M:18; F:32 Age (years): G1: 45.5 (12.2) G2:48.2 (13.2)	G1: GRP (*n* = 42) G2: stabilization exercise (*n* = 36) Drop out *n* = 22	- Baseline (T0) - 3 months (T1) - 6 months (T2) Follow-up, 4 months after treatment	RMDQ VAS ODI FFT	G1 vs. G2 RMDQ, *p* < 0.001 VAS, *p* < 0.001 ODI, *p* = 0.003 FFT, *p* = 0.008
Guimarães, ML. et al., 2013 [19] To evaluate the quality of life (QL) with the use of the SF-36 questionnaire in patients with chronic nonspecific low back pain (CNLBP).	*n* = 30 Gender G1: M:4; F: 6 G2: M:3; F:7 G3: M:3; F:7	G1: isostretching (*n* = 10)G2: GRP (*n* = 10) G3: isostretching + GPR (*n* = 10)	- Baseline (T0) - 3 months (T1) Follow-up, 2 months after treatment	VASSF-36	VAS G1: N/A; G2: *p* < 0.001; G: N/A SF36 G1: N/A; G2: *p* < 0.001; G: N/A
Castagnoli, C. et al., 2015 [20] Comparing global postural re-education (GPR) to a standard physiotherapy treatment (PT) based on active exercises, stretching and massaging to improve pain and function in chronic low back pain (CLBP) patients.	*n* = 79 Gender G1: M:4; F: 26 G2: M:7; F:23 Age (years): G1: 58.97 (0.44) G2: 62.54 (13.19)	G1: GRP (*n* = 30) G2: protocol treatment (*n* = 30) Drop out *n* = 19	- Baseline (T0) - Discharge, after 15 sessions (approx. 2 months) (T1) Follow-up 12 months after treatment (T2)	RMDQNRS	RMDQ G1(T0 vs. T1) *p* < 0.00; G1 (T0 vs. T2) *p* = 0.24 G2(T0 vs. T1) *p* < 0.00; G2 (T0 vs. T2) *p* = 0.12 NRSG1(T0 vs. T1) *p* < 0.00; G1 (T0 vs. T2) *p* = 0.02 G2(T0 vs. T1) *p* < 0.00; G2 (T0 vs. T2) *p* = 0.12
Lawand, P. et al., 2015 [23] To assess to assess the effect of a muscle stretching program using the GPR method in pain, function, quality of life and depressive symptoms in patients with chronic low back pain.	*n* = 61 Gender G1: M:6; F: 25 G2: M:8; F:22 Age (years): G1: 49.4 (12.0) G2: 47.5 (11.9)	G1: GRP (*n* = 31) G2: drug treatment (*n* = 30)	- Baseline (T0) - 3 months (T1) Follow-up, 6 months (T2)	RMQ VAS SF-36 Beck Inventory	RMQ G1 *p* < 0.001; G2 *p* = 0.264; G1 vs. G2 *p* < 0.001 VASG1 *p* < 0.001; G2 *p* = 0.340; G1 vs. G2 *p* < 0.001 SF36 G1 vs. G2 Functional capacity *p* = 0.396 Limitation in physical aspects *p* = 0.040 Pain *p* = 0.047 General health *p* = 0.363 Vitality *p* = 0.003 Social aspects *p* = 0.103 Emotional aspects *p* = 0.008 Mental health *p* = 0.034 Beck Inventory (no differences)
Soares, P. et al., 2015 [22] To compare the effects of the school of posture program (PEP) and global postural re-education (RPG) on pain levels and range of motion in patients with chronic low back pain.	*n* = 30 Gender N/A Age (years): G1: 46.3 (8.5) G2: 43.6 (10.93) G3: 44.30 (10.68)	G1: GRP (*n* = 10) G2: back school exercises and muscle strengthening (*n* = 10) G3: control (*n* = 10)	- Baseline (T0) - 3 months (T1)	Borg scale CR10 Goniometry	Borg Scale G1 vs. G3, *p* < 0.0001 G2 vs. G3, *p* < 0.0001 G1 vs. G2, *p* > 0.05 Hip extension G1 vs. G3, *p* = 0.019 G2 vs. G3, *p* = 0.006 G1 vs. G2, *p* > 0.05 Lumbar spine flexion G1 vs. G3, *p* = 0.020 G2 vs. G3, *p* = 0.018 G1 vs. G2, *p* > 0.05 Reduction of pain scores in back
Moreschi, F. et al., 2016 [9] To analyze changes in muscle strength, flexibility, function and pain in patients with chronic low back pain who underwent isostretching and global posture re-education (GPR).	*n* = 43 Gender G1: M:5; F:16 G2: M:3; F: 15 Age (years): G1: 50.50 G2: 52	G1: GRP (*n* = 21) G2: isostretching (*n* = 18) Drop out *n* = 4	- Baseline (T0) - 1.5 months (T1)	RMQ VAS Sit and reach test Dynamometry postural analysis	RMQ G1 *p* = 0.000; G2 *p* = 0.000; G1 vs. G2 *p* = 0.192 VAS G1 *p* = 0.001; G2 *p* = 0.000; G1 vs. G2 *p* = 0.494 Sit and reach test G1 *p* = 0.006; G2 *p* = 0.039; G1 vs. G2 *p* > 0.15 Dynamometry G1 *p* = 0.002; G2 *p* = 0.000; G1 vs. G2 *p* > 0.15 Postural analysis G1 *p* = 0.001; G2 *p* = 0.007; *p* > 0.15
Longo, C. et al., 2016 [21] To investigate whether the standing posture with flexion of the trunk added to a standard group physical therapy may increase the LM thickness (primary aim) and reduce pain and disability (secondary aim) in patients with chronic non-specific LBP.	*n* = 14 Gender G1: M:2; F:5 G2: M:2; F: 5 Age (years): G1: 54.57 (8.16) G2: 49.14 (9.92)	G1: GRP (*n* = 7) G2: standard protocol (*n* = 7)	- Baseline (T0) - 1 month (T1) - 2 months (T2)	RMQ NRS Ultrasound examination	RMQ (T1-T0) G1 vs. G2 *p* = 0.018 (T2-T0) G1 vs. G2 *p* = 0.042 NRS (T1-T0) G1 vs. G2 *p* = 0.071 (T2-T0) G1 vs, G2 *p* = 0.891 Ultrasound examination (T1-T0) G1 vs. G2 *p* > 0.05

**Table 2 jcm-10-05327-t002:** Begg and Egger tests.

Variable	Begg (*p*)	Egger (*p*)
RMDQ	*p* = 0.3082	*p* = 0.3670
ODI	*p* = 1.0000	
VAS	*p* = 0.2963	*p* = 0.3247
NRS	*p* = 1.000	
VAS + NRS	*p* = 0.8065	*p* = 0.5766

**Table 3 jcm-10-05327-t003:** Evaluation of the quality of the studies according to the PEDro scale.

Evaluation Criteria (Items)	1	2	3	4	5	6	7	8	9	10	11	Total Score
Author, Year												
Bonetti, F. et al., 2010	1	0	0	1	0	0	0	1	1	1	1	6
Guimarães, M.L. et al., 2013	1	1	0	0	0	0	0	0	0	1	0	2
Castagnoli, C. et al., 2015	1	0	0	1	0	0	0	0	0	1	1	4
Lawand, P. et al., 2015	1	1	1	1	0	0	1	1	1	1	1	8
Soares, P. et al., 2015	1	1	1	1	0	0	0	0	0	1	0	4
Moreschi, F.A. et al., 2016	1	1	1	1	0	0	1	1	0	1	1	7
Longo, C. et al., 2016	1	1	0	1	0	0	0	1	1	1	1	6
Pecorone, F. et al., 2020	1	1	0	1	0	0	0	1	0	1	1	5

Score 0: the criterion is not met. Score 1: the criterion is met. 1. Eligibility criteria were specified (This item is not used to calculate the PEDro score). 2. Subjects were randomly allocated to groups. 3. Allocation was concealed. 4. The groups were similar at baseline regarding the most important prognostic indicators. 5. There was blinding of all subjects. 6. There was blinding of all therapists who administered the therapy. 7. There was blinding of all assessors who measured at least one key outcome. 8. Measures of at least one key outcome were obtained from more than 85% of the subjects initially allocated to groups. 9. All subjects for whom outcome measures were available received the treatment or control condition as allocated, or, where this was not the case, data for at least on key outcome were analyzed by “intention to treat”. 10. The results of between-group statistical comparisons are reported for at least one key outcome. 11. The study provides both point measures and measures of variability for at least one key outcome.

**Table 4 jcm-10-05327-t004:** Quality of the evidence.

Certainty Assessment							No. of Patients		Effect		Certainty	Importance
No. of Studies	Study Design	Risk of Bias	Inconsistency	Indirectness	Imprecision	Other Considerations	RPG	Placebo	Relative (95% CI)	Absolute (95% CI)		
RMDQ (follow-up: range 2 months to 7 months; scale: from 0 to 24)												
4	randomized trials	not serious	not serious	not serious	not serious	none	109	103	-	SMD 0.55 lower (0.83 lower to 0.27 lower)	⨁⨁⨁⨁ HIGH	IMPORTANT
ODI												
2	randomized trials	not serious	not serious	not serious	not serious	none	72	66	-	SMD 0.48 lower (0.82 lower to 0.14 lower)	⨁⨁⨁⨁ HIGH	IMPORTANT
VAS												
3	randomized trials	not serious	not serious	not serious	serious	strong association	82	76	-	SMD 0.69 lower (1.01 lower to 0.37 lower)	⨁⨁⨁⨁ HIGH	IMPORTANT
NSR												
3	randomized trials	not serious	not serious	not serious	not serious	none	58	55	-	SMD 0.49 lower (0.87 lower to 0.12 lower)	⨁⨁⨁⨁ HIGH	NO IMPORTANT
VAS + NRS												
5	randomized trials	not serious	not serious	not serious	not serious	none	119	113	-	MD 1.32 lower (1.87 lower to 0.77 lower)	⨁⨁⨁⨁ HIGH	IMPORTANT
SF36												
3	randomized trials	not serious	not serious	not serious	not serious	none	50/50 (100.0%)	50/50 (100.0%)	not estimable		⨁⨁⨁⨁ HIGH	NO IMPORTANT

⨁ The number of symbols indicates the degree of certainty.

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
