# Peer review of "Effectiveness of Global Postural Re-Education in Chronic Non-Specific Low Back Pain: Systematic Review and Meta-Analysis"

_jcm, 2021, doi:10.3390/jcm10225327_

Round 1
Reviewer 1 Report
Review of manuscript “Systematic review and meta‐analysis Effectiveness of Global Postural Reeducation in non‐specific Low Back Pain”
Thank you for the possibility to review this meta-analysis regardin important problem – nonspecific low back pain.
A systematic review and meta‐analysis was carried out using PRISMA2020.
Randomized controlled trials that analyzed pain and patient‐reported outcomes were included in this review. 4 meta‐analyses were performer. This meta‐analysis provides reliable evidence that GPR may be an effective method for treating LBP by decreasing pain and improving function, with high evidence.
Concerns:
Papers included in systematic review analyzed chronic LBP – why in this manuscript “chronic” was not mentioned?
Authors wrongly wrote “It is the most common musculoskeletal complaint in the emergency department and 50% of these patients require hospitalization.[13]”
In the cited article - The Emergent Evaluation and Treatment of Neck and Back Pain -it is written “In the ED, approximately 1 in 50 patients with back pain require hospitalization” it is 2% not 50%
Why “Pelvic Girdle Pain” was included in searching. It is different disorders, not related to nonspecific LBP. This same regards Urogenital System and Male Urogenital Diseasesʺ[Mesh] OR ʺFemale Urogenital Diseasesʺ[Mesh] OR ʺUrogenital Surgical Proceduresʺ[Mesh] OR ʺPelvic Organ Prolapseʺ[Mesh]) – they are different disorders. In my opinion it is methodological mistake.
Minor: citations should be before dot. In some cases are after.
Line 53 there is “ .[3,11]3,4” ?
Author Response
Reviewer 1:
RV: Review of manuscript “Systematic review and meta‐analysis Effectiveness of Global Postural Reeducation in non‐specific Low Back Pain”
Thank you for the possibility to review this meta-analysis regarding important problem – nonspecific low back pain.
A systematic review and meta‐analysis was carried out using PRISMA2020.
Randomized controlled trials that analyzed pain and patient‐reported outcomes were included in this review. 4 meta‐analyses were performer. This meta‐analysis provides reliable evidence that GPR may be an effective method for treating LBP by decreasing pain and improving function, with high evidence.
Concerns:
RV: Papers included in systematic review analyzed chronic LBP – why in this manuscript “chronic” was not mentioned?
AA: Thank you very much to the reviewer at this point. We fully agree with the reviewer on this point. The term "Chronic" has been included throughout the manuscript.
RV: Authors wrongly wrote “It is the most common musculoskeletal complaint in the emergency department and 50% of these patients require hospitalization [13].” In the cited article [13] - The Emergent Evaluation and Treatment of Neck and Back Pain -it is written “In the ED, approximately 1 in 50 patients with back pain require hospitalization” it is 2% not 50%
AA: Thank you for your valuable comment at this point. The mistake has been modified according to the correct information in the reference number 13.
RV: Why “Pelvic Girdle Pain” was included in searching. It is different disorders, not related to nonspecific LBP. This same regards Urogenital System and Male Urogenital Diseasesʺ [Mesh] OR ʺFemale Urogenital Diseasesʺ [Mesh] OR ʺUrogenital Surgical Proceduresʺ [Mesh] OR ʺPelvic Organ Prolapseʺ [Mesh]) – they are different disorders. In my opinion it is methodological mistake.
AA: Thank you for your valuable comment that helped us to improve the quality of the paper. We fully agree with the reviewer on this point. These disorders were included in the search strategies because in many cases they are considered as nonspecific low back pain. However, the articles that were on topics that were not really chronic non-specific low back pain were ultimately excluded from this review.
RV: Minor: citations should be before dot. In some cases, are after.
AA: Thank you for pointing this. All citations have been doble checked, and now the citations are in the correct position.
RV: Line 53 there is “.[3,11]3,4” ?
AA: Thank you for pointing this too. In the same sense as the previous comment, the citations have been double checked. The mistake has been modified according to the correct information.
Reviewer 2 Report
Dear authors,
The manuscript entitles "Effectiveness of Global Postural Reeducation in non-specific Low Back Pain: Systematic review and meta-analysis" has been reviewed. The main topic has already been covered by other articles. The written English needs great revisions as well.
Author Response
Dear authors,
RV: The manuscript entitles "Effectiveness of Global Postural Reeducation in non-specific Low Back Pain: Systematic review and meta-analysis" has been reviewed. The main topic has already been covered by other articles.
AA: Thank you for your valuable comment at this point. The reviewer remarks at this point that there is evidence on the topic of this study. From our previous feasibility study of this work, only one evidence was found with similarity to the present study:
https://pubmed.ncbi.nlm.nih.gov/?term=global+postural+reeducation+AND+Low+back+pain&filter=pubt.systematicreview
This one work is from our collages in the University of Jaén (Rafael Lomas et al. 2017). The aim of this study was to investigate the effects of global postural re-education (GPR) on the treatment of spinal disorders by performing a systematic review and a meta-analysis. However, in this review authors included in the meta-analysis in different disorders, with differences in the management of these pathologies (ankylosing spondylitis, neck pain and LBP). All meta-analysis from this review, shown the results with these three pathologies together.
In this way, one recent study from Corp et al. 2021 and some other authors from recent pass, concluded that is a notable difference between back and neck pain management, for example analgesics for neck pain (not for back pain); or options for back pain‐specific subgroups work‐based interventions. Therefore, the management of these pathologies is different, and we justified the meta-analysis only for chronic non-specific LBP in this way.
Reference: Corp N, Mansell G, Stynes S, Wynne-Jones G, Morsø L, Hill JC, van der Windt DA. Evidence-based treatment recommendations for neck and low back pain across Europe: A systematic review of guidelines. Eur J Pain. 2021 Feb;25(2):275-295. doi: 10.1002/ejp.1679. Epub 2020 Nov 12. PMID: 33064878; PMCID: PMC7839780.
RV: The written English needs great revisions as well.
AA: Thank you for your valuable comment that helped us to improve the quality of the paper. We will send to the English Editing Service from MPDI according to the reviewer 3 recommendation.
Reviewer 3 Report
It is a well-written systematic review regarding the effectiveness of global postural reeducation on low back pain. This study included seven trials and performed three meta-analyses on three aspects: pain, disability and quality of life. Overall, it shows that reeducation of posture decreases the perception of pain and improves the daily function/quality of life.
Minor revision:
As indicated by authors, the quality of the studies included in the meta-analysis is in general average. This is partially due to the difficulty of blinding the subjects and therapist. Therefore, the perceived improvement in pain/disability might largely due to placebo effects. It is much appreciated that authors of this study acknowledged the limitation and used objective quality control scale (PEDro scale) to screen the quality of studies. As illustrated by table3, study of Guimaraes, et al 2013 only scored 2 indicating low quality. It might be necessary to exclude this study from meta-analysis unless other compelling reasons suggest otherwise.
